fluid mechanics/biomathematics/computational mathematics

Stokes flow, stokeslets, biological fluid dynamics

**Author for correspondence:**
D. J. Smith
e-mail: d.j.smith@bham.ac.uk

# The art of coarse Stokes: Richardson extrapolation improves the accuracy and efficiency of the method of regularized stokeslets

## M. T. Gallagher[1] and D. J. Smith[2]

[1]Centre for Systems Modelling and Quantitative Biomedicine, and [2]School of Mathematics, University of Birmingham, Birmingham, UK

MTG, 0000-0002-6512-4472; DJS, 0000-0002-3427-0936

The method of regularized stokeslets is widely used in microscale biological fluid dynamics due to its ease of implementation, natural treatment of complex moving geometries, and removal of singular functions to integrate. The standard implementation of the method is subject to high computational cost due to the coupling of the linear system size to the numerical resolution required to resolve the rapidly varying regularized stokeslet kernel. Here, we show how Richardson extrapolation with coarse values of the regularization parameter is ideally suited to reduce the quadrature error, hence dramatically reducing the storage and solution costs without loss of accuracy. Numerical experiments on the resistance and mobility problems in Stokes flow support the analysis, confirming several orders of magnitude improvement in accuracy and/or efficiency.

# 1. Introduction: the method of regularized stokeslets

Flow problems associated with flagellar propulsion of cells, cilia-driven fluid transport, and synthetic microswimmers, are characterized by the inertialess regime of approximately zero Reynolds number flow, described mathematically—in Newtonian flow—by the Stokes flow equations,

$$-\boldsymbol{\nabla} p + \mu \nabla^2 \boldsymbol{u} = \mathbf{0}, \quad \boldsymbol{\nabla} \cdot \boldsymbol{u} = 0. \tag{1.1}$$

Typically, these conditions are associated with no-flux, no-penetration conditions on complex-shaped moving boundaries modelling cell surfaces and motile appendages. For a detailed

introduction to the subject, see the recent text [1]. A range of mathematical and computational techniques are available to approach this problem; a computational method that has seen significant uptake and development over the last two decades is the method of regularized stokeslets, first described by Cortez [2] and subsequently elaborated for three-dimensional flow [3,4].

This technique can be viewed as a modification of the method of fundamental solutions and/or the boundary integral method for Stokes flow [5], the basis for which is the *stokeslet* [6] or *Oseen tensor* [7]:

$$S_{jk}(\pmb{x}, \pmb{y}) = \frac{\delta_{jk}}{|\pmb{x} - \pmb{y}|} + \frac{(x_j - y_j)(x_k - y_k)}{|\pmb{x} - \pmb{y}|^3} \tag{1.2}$$

and

$$P_k(\pmb{x}, \pmb{y}) = \frac{2(x_k - y_k)}{|\pmb{x} - \pmb{y}|^3}. \tag{1.3}$$

The pair of tensors $S_{jk}$, $P_k$ provide the solutions $\pmb{u} = (8\pi\mu)^{-1}(S_{1k}, S_{2k}, S_{3k})$ and $p = (8\pi)^{-1}P_k$ to the singularly forced Stokes flow equations,

$$-\pmb{\nabla} p + \mu\nabla^2\pmb{u} + \delta(\pmb{x} - \pmb{y})\hat{\pmb{e}}_k = \pmb{0} \tag{1.4}$$

and

$$\pmb{\nabla} \cdot \pmb{u} = 0, \tag{1.5}$$

where $\delta(\pmb{x})$ is the three-dimensional Dirac delta distribution and $\hat{\pmb{e}}_k$ is a unit basis vector pointing in the $k$-direction. Equations (1.2) and (1.3) are singular when the source point $\pmb{x}$ and field point $\pmb{y}$ coincide. To facilitate numerical computation, the method of regularized stokeslets instead considers the Stokes flow equation with spatially smoothed point force,

$$-\pmb{\nabla} p + \mu\nabla^2\pmb{u} + \phi_\epsilon(\pmb{x} - \pmb{y})\hat{\pmb{e}}_k = \pmb{0} \tag{1.6}$$

and

$$\pmb{\nabla} \cdot \pmb{u} = 0, \tag{1.7}$$

where $\phi_\epsilon(\pmb{x})$ is a family of 'blob' functions approximating $\delta(\pmb{x})$ as $\epsilon \to 0$.

Several different choices for $\phi_\epsilon$ and associated regularized stokeslets $S_{jk}^\epsilon$ have been studied; the most extensively used was presented in the original three-dimensional formulation of Cortez *et al.* [3],

$$\phi_\epsilon(\pmb{x}) = \frac{15\epsilon^4}{8\pi(|\pmb{x}|^2 + \epsilon^2)^{7/2}}, \tag{1.8}$$

$$P_k^\epsilon(\pmb{x}, \pmb{y}) = \frac{x_k}{(|\pmb{x}|^2 + \epsilon^2)^{5/2}}(2|\pmb{x}|^2 + 5\epsilon^2) \tag{1.9}$$

and

$$S_{jk}^\epsilon(\pmb{x}, \pmb{y}) = \frac{\delta_{jk}(|\pmb{x}|^2 + 2\epsilon^2) + x_j x_k}{(|\pmb{x}|^2 + \epsilon^2)^{3/2}}. \tag{1.10}$$

Developments focusing on the use of alternative blob functions to improve convergence include [8] (near-field) and, more recently, [9] (far-field).

The pressure $P_k^\epsilon(\pmb{x}, \pmb{y}) \sim P_k(\pmb{x}, \pmb{y})$ and velocity $S_{jk}^\epsilon(\pmb{x}, \pmb{y}) \sim S_{jk}(\pmb{x}, \pmb{y})$ as $\epsilon \to 0$; moreover the corresponding single layer boundary integral equation is

$$u_j(\pmb{x}) = -\frac{1}{8\pi\mu}\iint_B S_{jk}^\epsilon(\pmb{x}, \pmb{y})f_k(\pmb{y})\,\mathrm{d}S_{\pmb{y}} + O(\epsilon^p), \tag{1.11}$$

where $p = 1$ for $\pmb{x}$ on or near $B$ and $p = 2$ otherwise [3]. In equation (1.11) and below, summation over repeated indices in $j = 1, 2, 3$ or $k = 1, 2, 3$ is implied. The reduction to the single-layer potential is discussed by e.g. [3,5,10]; in brief, this equation can describe flow due to motion of a rigid body, or with suitable adjustment to $f_k$, the flow exterior to a body which does not change volume. A feature common to both standard and regularized stokeslet versions of the boundary integral equation is non-uniqueness of the solution $f_k$. This non-uniqueness occurs due to incompressibility of the stokeslet, i.e. provided the interior of $B$ maintains its volume, then $\iint_B S_{jk}n_k\,\mathrm{d}S_{\pmb{y}} = 0$ so that if $f_k$ is a solution of equation (1.11) then so is $f_k + an_k$ for any constant $a$. From the perspective of the original partial differential equation system, the non-uniqueness follows from the fact that the pressure part of the solution to equations (1.1) with velocity-only boundary conditions is determined only up to an

additive constant. This issue is not dynamically important, and moreover the discretized approximations to the system described below result in invertible matrices.

Boundary integral methods have the major advantage of removing the need for a volumetric mesh, which both reduces computational cost, and moreover avoids the need for complex meshing and mesh movement. The key strength of the method of regularized stokeslets is in enabling the boundary integral method to be implemented in a particularly simple way: by replacing the integral by a numerical quadrature rule $\{\boldsymbol{x}[n], w[n], \mathrm{d}S(\boldsymbol{x}[n])\}$ (abscissae, weight and surface metric), equation (1.11) may be approximated by,

$$u_j(\boldsymbol{x}[m]) \approx \frac{1}{8\pi\mu} \sum_{n=1}^{N} S_{jk}^{\epsilon}(\boldsymbol{x}[m], \boldsymbol{x}[n]) f_k(\boldsymbol{x}[n]) w[n] \, \mathrm{d}S(\boldsymbol{x}[n]). \tag{1.12}$$

As is standard terminology in numerical methods for integral equations, we will refer to this as the *Nyström* discretization [11]. By allowing $m = 1, \ldots, N$ and $j = 1, 2, 3$, a dense system of $3N$ linear equations in $3N$ unknowns $F_k[n] := f_k(\boldsymbol{x}[n]) w[n] \, \mathrm{d}S(\boldsymbol{x}[n])$ is formed. The diagonal entries when $j = k$ and $m = n$ are finite but numerically on the order of $1/\epsilon$, leading to (by the Gershgorin circle theorem) a well-conditioned matrix system.

The approach outlined above can be used to solve the *resistance problem* in Stokes flow, which involves prescribing a rigid body motion and calculating the force distribution, and hence total force and moment on the body. Once the force and moment associated with each of the six rigid body modes (unit velocity translation in the $x_j$ direction, unit angular velocity rotation about $x_j$ axis, for $j = 1, 2, 3$) are calculated, the *grand resistance matrix A* can be formed [5], which by linearity of the Stokes flow equations relates the force $\boldsymbol{F}$ and moment $\boldsymbol{M}$ to the velocity $\boldsymbol{U}$ and angular velocity $\boldsymbol{\Omega}$ for any rigid body motion;

$$\begin{pmatrix} \boldsymbol{F} \\ \boldsymbol{M} \end{pmatrix} = \underbrace{\begin{pmatrix} A_{FU} & A_{F\Omega} \\ A_{MU} & A_{M\Omega} \end{pmatrix}}_{A} \begin{pmatrix} \boldsymbol{U} \\ \boldsymbol{\Omega} \end{pmatrix}. \tag{1.13}$$

For example, for a sphere of radius $a$ centred at the origin, the matrix blocks are $A_{FU} = 6\pi\mu a I$, $A_{F\Omega} = 0 = A_{MU}$ and $A_{M\Omega} = 8\pi\mu a^3 I$ where $I$ is the $3 \times 3$ identity matrix.

A closely related problem is the two-step calculation of the flow field due to a prescribed boundary motion; starting with prescribed surface velocities $u_j(\boldsymbol{x}[m])$, first, the discrete force distribution $F_k[n]$ is found by inversion of the Nyström matrix system; the velocity field at any point in the fluid $\tilde{\boldsymbol{x}}$ can then be found through the summation,

$$u_j(\tilde{\boldsymbol{x}}) = \frac{1}{8\pi\mu} \sum_{n=1}^{N} S_{jk}^{\epsilon}(\tilde{\boldsymbol{x}}, \boldsymbol{x}[n]) F_k[n]. \tag{1.14}$$

The *mobility problem* is formulated by prescribing the total force and moment on the body (yielding six scalar equations) and augmenting the system with unknown velocity $\boldsymbol{U}$ and angular velocity $\boldsymbol{\Omega}$, which adds six scalar unknowns, so that a $(3N + 6) \times (3N + 6)$ system is formed. At a given time, these unknowns can be related to the evolution of the body trajectories (in terms of position $\boldsymbol{x}_0$ and two basis vectors $\boldsymbol{b}^{(1)}$ and $\boldsymbol{b}^{(2)}$), through a system of nine ordinary differential equations

$$\dot{\boldsymbol{x}}_0 = \boldsymbol{U}(\boldsymbol{x}_0, \boldsymbol{b}^{(1)}, \boldsymbol{b}^{(2)}, t), \quad \dot{\boldsymbol{b}}^{(j)} = \boldsymbol{\Omega}(\boldsymbol{x}_0, \boldsymbol{b}^{(1)}, \boldsymbol{b}^{(2)}, t) \times \boldsymbol{b}^{(j)}, \quad j = 1, 2, \tag{1.15}$$

which can be solved using available packages such as MATLAB's ode45.

Finally, the *swimming problem* further prescribes the motion of cilia or flagella with respect to a body frame (typically, a frame in which the cell body is stationary), and often assumes zero total force and moment (neglecting gravity and other forces such as charge), again resulting in a $(3N + 6) \times (3N + 6)$ system. The key numerical features and challenges of the method of regularized stokeslets are exhibited by the resistance and mobility problems, which will therefore be our primary focus.

## 2. Convergence properties of the Nyström discretization

Equation (1.12) is subject to the $O(\epsilon)$ regularization error in the boundary integral equation, and the discretization error in the approximation of the integral. The integrand consists of a product: the slowly varying traction $f_k(\boldsymbol{y})$ and the stokeslet kernel $S_{jk}^{\epsilon}(\boldsymbol{x}[m], \boldsymbol{y})$ which is rapidly varying when $\boldsymbol{y} \approx \boldsymbol{x}[m]$. The error associated with discretization of the traction is at worst $O(h)$, where $h$ is the

characteristic spacing between points. The dominant error for the stokeslet kernel can be shown to be

$$O(\epsilon^{-1}h^2), \tag{2.1}$$

(see [12], *contained case*, equation (2.7)).

Reducing the $O(\epsilon)$ regularization error by reducing $\epsilon$ therefore increases the $O(\epsilon^{-1}h^2)$ stokeslet quadrature error, necessitating refinement of the discretization length $h$. To reduce $\epsilon$ by a factor of $R$ requires indicatively reducing $h$ by a factor of $\sqrt{R}$, hence increasing the number of surface points and therefore degrees of freedom $N$ by a factor of $R$. The cost of assembling the dense linear system then increases by a factor of $R^2$, and the cost of a direct linear solver by a factor of $R^3$. This calculation shows that, for example, improving from a 10% relative error to a 1% relative error may indicatively incur a cost increase of 1000 times. There are several approaches available already to address this issue, which involve a range of computational complexities: the fast multipole method [13], boundary element regularized stokeslet method [14] and the nearest-neighbour discretization [15], for example. In the next section, we will describe and analyse a very simple technique which alone, or potentially in combination with the above, improves the order of the regularization error, thereby enabling a coarser $\epsilon$ and hence alleviating the quadrature error. We will then briefly review an alternative 'coarse' approach, the nearest-neighbour method, a benchmark with similar implementational simplicity. Numerical experiments will be shown in the Results (§5), and we close with brief Discussion (§6).

## 3. Richardson extrapolation in regularization error

Consider the approximation of a physical quantity (e.g. moment on a rotating body) which has exact value $M^*$. The value of this quantity calculated with discretization of size $h$ and regularization parameter $\epsilon$ is denoted

$$M(\epsilon, h) = M^* + E_r(\epsilon) + E_d(h; \epsilon), \tag{3.1}$$

where $E_r(\epsilon)$ is the regularization error associated with the (undiscretized) integral equation, and $E_d(h; \epsilon)$ is the discretization error, which as indicated also has an indirect dependence on $\epsilon$ via the quadrature.

Recall that

$$E_r(\epsilon) = O(\epsilon) \tag{3.2}$$

and

$$E_d(h; \epsilon) = E_f(h) + E_q(h; \epsilon) = O(h) + O\left(\frac{h^2}{\epsilon}\right), \tag{3.3}$$

where $E_f(h)$ is the error associated with the force discretization and $E_q(h; \epsilon)$ is the quadrature error. The analysis below will focus on the situation in which the regularization parameter $\epsilon$ is not excessively small, so that the quadrature error $(h^2/\epsilon)$ is subleading and hence the discretization error has minimal dependence on $\epsilon$, thus $E_d(h; \epsilon) \approx E_d(h; \epsilon_0)$ for some representative value $\epsilon_0$. Writing

$$M(\epsilon; h) = M^* + E_r(\epsilon) + E_d(h; \epsilon_0), \tag{3.4}$$

we may then expand,

$$M(\epsilon; h) = M^* + \epsilon E_r'(0) + \frac{\epsilon^2}{2}E_r''(0) + O(\epsilon^3) + E_d(h; \epsilon_0). \tag{3.5}$$

Evaluation of $M(\epsilon_\ell, h)$ for three values of $\epsilon_\ell$ in this range results in a linear system,

$$\begin{pmatrix} M(\epsilon_1, h) \\ M(\epsilon_2, h) \\ M(\epsilon_3, h) \end{pmatrix} = \underbrace{\begin{pmatrix} 1 & \epsilon_1 & \epsilon_1^2 \\ 1 & \epsilon_2 & \epsilon_2^2 \\ 1 & \epsilon_3 & \epsilon_3^2 \end{pmatrix}}_{B} \begin{pmatrix} M^* \\ E_r'(0) \\ E_r''(0)/2 \end{pmatrix} + \begin{pmatrix} E_d(h; \epsilon_0) + O(\epsilon_1^3) \\ E_d(h; \epsilon_0) + O(\epsilon_2^3) \\ E_d(h; \epsilon_0) + O(\epsilon_3^3) \end{pmatrix}. \tag{3.6}$$

Applying the matrix inverse,

$$B^{-1}\begin{pmatrix} M(\epsilon_1, h) \\ M(\epsilon_2, h) \\ M(\epsilon_3, h) \end{pmatrix} = \begin{pmatrix} M^* \\ E_r'(0) \\ E_r''(0)/2 \end{pmatrix} + B^{-1}\begin{pmatrix} E_d(h; \epsilon_0) + O(\epsilon_1^3) \\ E_d(h; \epsilon_0) + O(\epsilon_2^3) \\ E_d(h; \epsilon_0) + O(\epsilon_3^3) \end{pmatrix}. \tag{3.7}$$

Hence, the estimate

$$\widetilde{M}(\epsilon_1, \epsilon_2, \epsilon_3; h) := \begin{pmatrix} 1 & 0 & 0 \end{pmatrix} B^{-1} \begin{pmatrix} M(\epsilon_1, h) \\ M(\epsilon_2, h) \\ M(\epsilon_3, h) \end{pmatrix} \tag{3.8}$$

provides an approximation to $M^*$ that has error

$$E_d(h; \epsilon_0) + O(\epsilon_1^3 + \epsilon_2^3 + \epsilon_3^3). \tag{3.9}$$

This improvement in order of accuracy comes at a small multiplicative cost associated with solving the problem three times; however, as these are three independent calculations they are ideally placed to exploit parallel computing architecture, thus reducing the additional computational cost.

# 4. Comparison with the nearest-neighbour regularized stokeslet method

Before carrying out numerical experiments, we will briefly recap a different strategy to address the $\epsilon$-dependence of the linear system size which we have developed and described recently, in order to provide a benchmark with similar implementational simplicity. The *nearest-neighbour* version of the regularized stokeslet method [16] aims to remove the $\epsilon$-dependence of the linear system size. This change is achieved by separating the degrees of freedom for traction from the quadrature by using two discretizations: a 'coarse force' set $\{\boldsymbol{x}[1], \ldots, \boldsymbol{x}[N]\}$ for the traction and a finer set $\{\boldsymbol{X}[1], \ldots, \boldsymbol{X}[Q]\}$ for the quadrature. If these sets are identical, the method reduces to the familiar Nyström discretization. In general, choosing $N < Q$ leverages the fact that the traction is more slowly varying than the near-field of the regularized stokeslet kernel. Discretizing the integral equation (1.11) on the fine set gives

$$u_j(\boldsymbol{x}[m]) = \sum_{q=1}^{Q} S_{jk}^{\epsilon}(\boldsymbol{x}[m], \boldsymbol{X}[q]) f_k(\boldsymbol{X}[q]) w[q] \, dS(\boldsymbol{X}[q]). \tag{4.1}$$

Based on the observation that the traction $f_k(\boldsymbol{X}[q])$ and associated weighting $w[q] dS(\boldsymbol{X}[q])$ are slowly varying, the method employs degrees of freedom $F_k[n]$ in the neighbourhood of each point of the coarse discretization, so that

$$w[q] \, dS(\boldsymbol{X}[q]) f_k(\boldsymbol{X}[q]) \approx \sum_{n=1}^{N} \nu[q, n] F_k[n], \tag{4.2}$$

where $\nu[q, n]$ is a sparse matrix defined so that $\nu[q, n] = 1$ if the closest coarse point to $\boldsymbol{X}[q]$ is $\boldsymbol{x}[n]$, and $\nu[q, n] = 0$ otherwise.

A detail that was not addressed in our recent papers ([15,17], for example) is that the closest coarse point to a given quadrature point may not be uniquely defined. Moreover, it is occasionally possible that, for sufficiently distorted discretizations, a coarse point may have no quadrature points associated with it at all, resulting in a singular matrix. In the former case, the weighting may be split between two or more coarse points, so that the sum of each row of $\nu[q, n]$ is still equal to 1. In the latter case, the coarse point may be removed from the problem, or (better) the quadrature discretization refined.

The approximation (4.2) leads to the linear system,

$$u_j(\boldsymbol{x}[m]) \approx \sum_{n=1}^{N} F_k[n] \sum_{q=1}^{Q} S_{jk}^{\epsilon}(\boldsymbol{x}[m], \boldsymbol{X}[q]) \nu[q, n]. \tag{4.3}$$

The computational complexity of the system is given by the $3N \times 3Q$ function evaluations required to assemble the stokeslet matrix, followed by the $O(N^3)$ solution of the dense linear system (for direct methods).

The nearest-neighbour method is subject to similar $O(\epsilon)$ regularization error and $O(h_f)$ discretization error (where $h_f$ is characteristic of the force point spacing) as the Nyström method. Analysis of the quadrature error associated with collocation [12] identifies two dominant contributions:

(i) *Contained case*: Quadrature centred about a force point which is also contained in the quadrature set is subject to a dominant error term $O(\epsilon^{-1} h_q^2)$, where $h_q$ is the spacing of the quadrature points; the Nyström method described above is a special case of this, with $h_q = h$;

(ii) *Disjoint case*: Quadrature centred about a force point which is not contained in the quadrature set is subject to a dominant error term $O(h_q/\delta)^2 \, h_q$, where $\delta > 0$ is the minimum distance between the force point and quadrature points. This term does not appear in the Nyström method error analysis. The term is written in this form because $\delta$ is typically similar in size to $h_q$ for a given quadrature set, so with a little care, $h_q/\delta$ behaves as a multiplicative constant.

For contained force and quadrature discretizations (i), the cost of quadrature is still an important consideration. Reducing $\epsilon$ by a factor of $R$, necessitates reducing $h_q^2$ by a factor of $R$, and hence increasing the number of quadrature points—and associated matrix assembly cost—by a factor of $R$. Therefore, any improvement to the order of convergence of the regularization error will result in a corresponding improvement in the reduction of quadrature error.

However, when disjoint force and quadrature discretizations (ii) are employed, the nearest-neighbour method is able to entirely decouple the strong dependence of the degrees of freedom (tied only to $h_f$) on the regularization parameter $\epsilon$ and quadrature discretization $h_q$. The nearest-neighbour method, therefore, provides a relatively efficient and accurate implementation of the regularized stokeslet method that, with minor care in the construction of the discretization sets, can be used as a benchmark. In the following section, we will assess the Richardson extrapolation approach against analytic solutions for two examples of the resistance problem, and against the nearest-neighbour method for an example mobility problem.

# 5. Results

We now turn our attention to the application of Richardson extrapolation to a series of model problems, comprising the calculation of:

(a) the grand resistance matrix for a unit sphere;
(b) the grand resistance matrix for a prolate spheroid; and
(c) the motion of a torus sedimenting under gravity.

For simulations (a) and (b), comparisons can be made to known exact solutions. For each test case (a–c), we compare the results of simulations using both the Nyström [Ny] and Nyström + Richardson [NyR] methods. For the latter, we choose extrapolation points $(\epsilon_1, \epsilon_2, \epsilon_3) = (\epsilon, \sqrt{2}\epsilon, 2\epsilon)$. The choice of extrapolation rule is discussed further in appendix A.

For each problem, we use the minimum distance between any two force points in the discretization as our comparative lengthscale $h$. For the [NyR] method, results are shown against the smallest value of the regularization parameter ($\epsilon_1$) used in the calculation. Simulations are performed with GPU acceleration (see [18]) using a Lenovo Thinkstation with an NVIDIA Quadro RTX 5000 GPU. Each of the test problems that we consider, however, are easily within the capabilities of more modest hardware.

## 5.1. The grand resistance matrix of a rigid sphere

Application of Stokes' Law gives the force exerted by the translation of the unit sphere with velocity $\boldsymbol{U} = (1, 0, 0)$ as $\boldsymbol{F} = (6\pi, 0, 0)$, and the moment exerted by the unit sphere with rotational velocity $\boldsymbol{\Omega} = (1, 0, 0)$ as $\boldsymbol{M} = (8\pi, 0, 0)$. From this, the grand resistance matrix $A$ can be constructed as in equation (1.13). We solve equation (1.12) [Ny] and equations (1.12) and (3.8) [NyR] for unit translations and rotations about each axis to obtain the numerical approximation to $A$, $A^\epsilon$. The relative error in the calculation is then given by the relation

$$\text{relative error} = \frac{\|A - A^\epsilon\|_2}{\|A\|_2}, \tag{5.1}$$

where $\|A\|_2$ denotes the 2-norm ($\|A\|_2 = \sup_{x \neq 0} \|Ax\|_2 / \|x\|_2$).

The unit sphere is discretized by projecting onto the six faces of a cube (figure 1*a*), with the number of scalar degrees of freedom (sDOF) shown plotted against the minimum spacing between points ($h$) in figure 1*b*. The relative error in calculating the grand resistance matrix as $h$ and $\epsilon$ are varied is shown in figure 1*c,d* ([Ny] and [NyR], respectively). We report results for an identical range of $\epsilon$ (and $h$) for both methods, although as described in §3, the [NyR] method is specifically designed to exploit larger values of $\epsilon$ for which the quadrature error is small, so the [NyR] results with $\epsilon = 0.1$–$0.4$ are most pertinent.

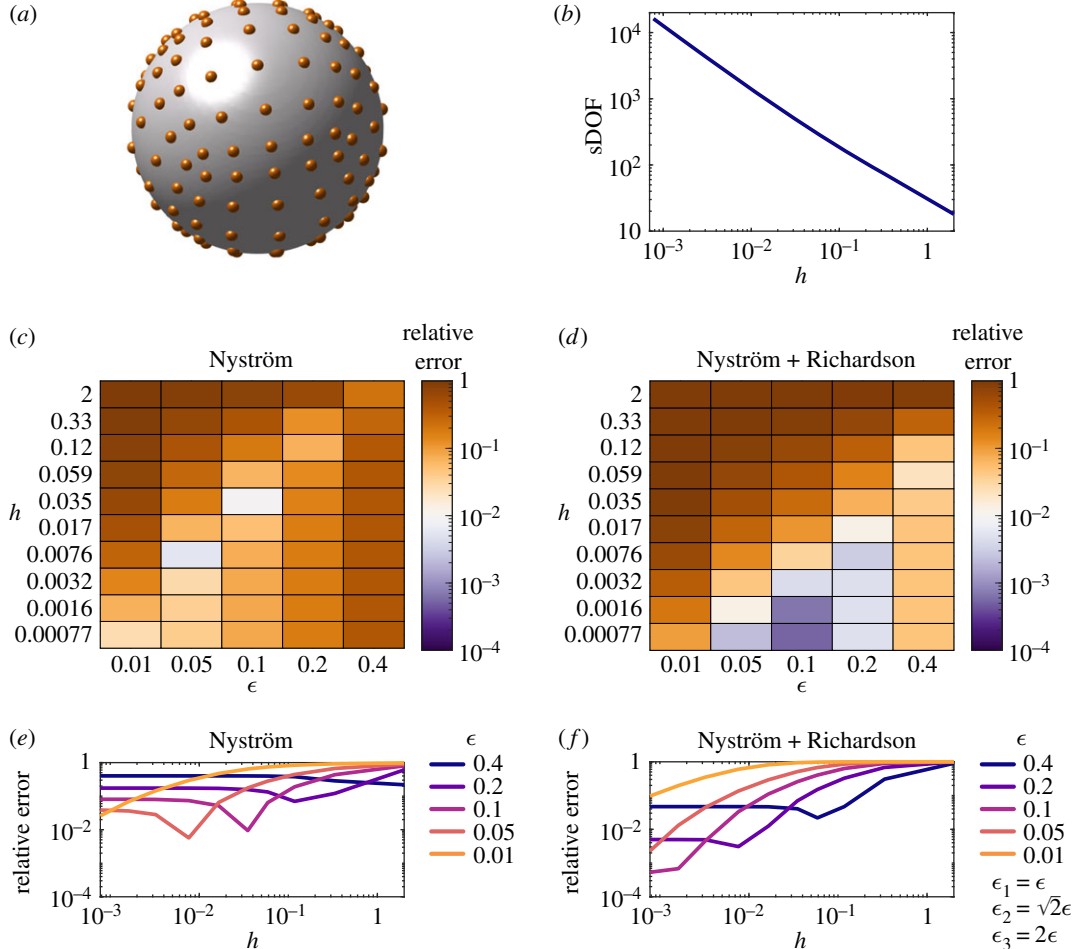

**Figure 1.** Relative error in calculating the grand resistance matrix for the unit sphere. (*a*) Sketch of the sphere discretization (orange dots). (*b*) The number of scalar degrees of freedom used in calculations as $h$ is varied. (*c*) and (*d*) The relative error of the Nyström and Nyström + Richardson methods as $\epsilon$ and $h$ are varied. (*e*) and (*f*) The same data plotted for each $\epsilon$ as $h$ is varied.

The [Ny] method is found to achieve 1% relative error for a select number of parameter pairs ($\epsilon$, $h$). This is strongly dependent, however, on the 'dip' in error which appears as $h$ is decreased for a given $\epsilon$ (evident in figure 1*e*) and is a consequence of the balance between the opposite-signed regularization and quadrature errors; the small $h$ plateau remains above 1% error for each choice of $\epsilon$. By contrast, the [NyR] method is able to significantly reduce the error in the plateau (figure 1*f*), resulting in sub-1% errors for $\epsilon$ as large as 0.2. Indeed with $\epsilon = 0.2$, the range of values of $h$ capable of producing acceptably accurate results extends from $h = 0.00077$ to $h = 0.0076$. As a result of the reduction in regularization error, brought about by the [NyR] extrapolation, this method is able to achieve a minimum relative error of 0.05% compared with 0.6% for the [Ny] method, and moreover, accurate performance no longer depends on a precise interplay between $h$ and $\epsilon$. In the simulations we performed, the [NyR] method was able to attain very accurate results (0.1% error) in 250 s of walltime.

## 5.2. The grand resistance matrix of a rigid prolate spheroid

To assess the performance on a system involving a modest disparity of length scales, the second model problem is the calculation of the grand resistance matrix for a prolate spheroid of major axis length 5 and minor axis length 1. Moreover, prolate spheroids are often used as models for both entire microscopic swimming cells, and for their propulsive cilia and flagella, and so provide an informative test geometry. The exact solution in the absence of other bodies is well known (e.g. [19]). Details of the discretization of the prolate spheroid are provided in appendix B.1. A sketch of the discretization and plot of sDOF as $h$ is varied are shown in figure 2*a*,*b*.

Similarly to the case of the unit sphere, the [Ny] method is able to achieve a minimum error of 0.8% for the smallest $\epsilon$ in this study and a specific choice of $h$ within the error dip (figure 2*c*,*e*). For each choice

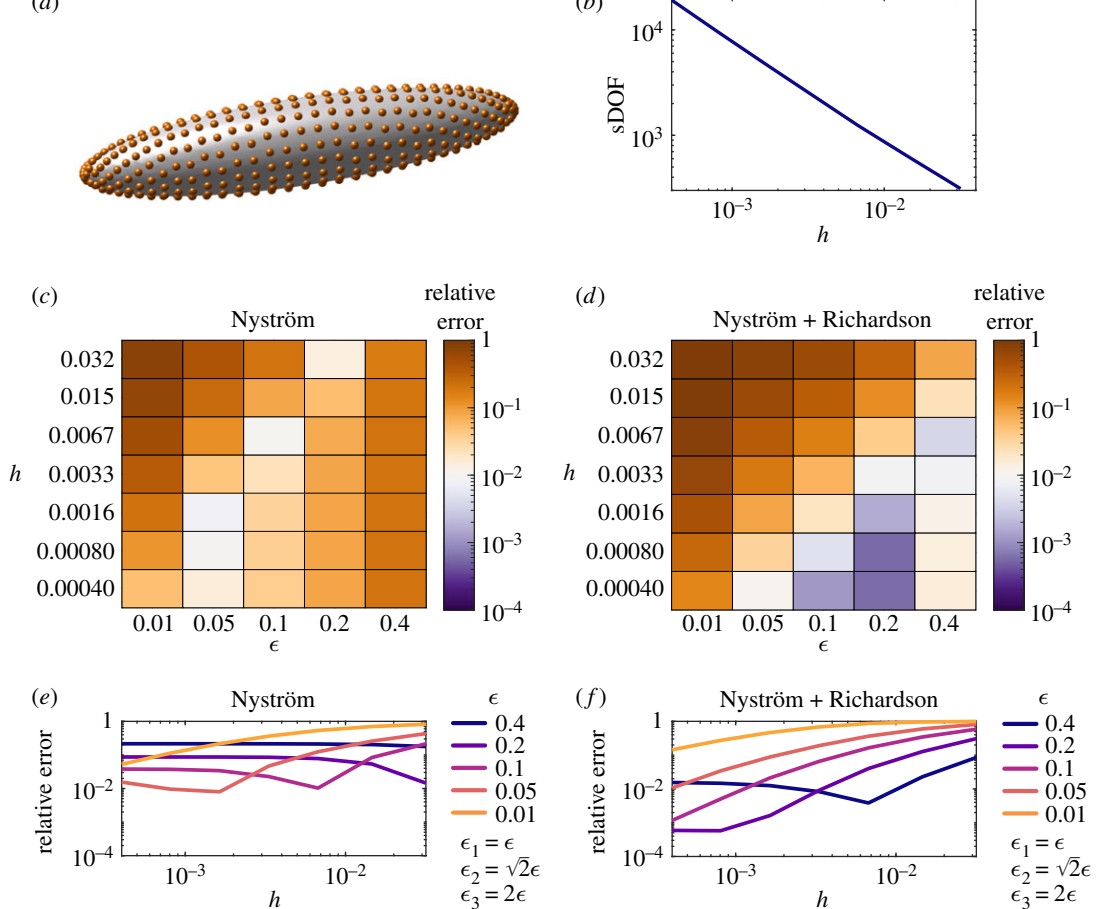

**Figure 2.** Relative error in calculating the grand resistance matrix for a prolate spheroid with major axis length $a = 5$ and minor axis length $c = 1$. (*a*) Sketch of the discretization (orange dots). (*b*) The number of scalar degrees of freedom used in calculations as $h$ is varied. (*c*) and (*d*) The relative error of the Nyström and Nyström + Richardson methods as $\epsilon$ and $h$ are varied. (*e*) and (*f*) The same data plotted for each $\epsilon$ as $h$ is varied.

of $\epsilon$, the error plateau for small $h$ is at least 1% relative error. Relatively large $\epsilon = 0.2$, 0.4 yield error plateaus of 8.7% and 22%, respectively.

The [NyR] method also exhibits this dip phenomenon; however, the reduction in regularization error provided by the Richardson extrapolation (figure 2*d,f*) results in significantly reduced error plateaus of 0.059% and 1.5% ($\epsilon = 0.2$ and 0.4, respectively), again being more robustly maintained over a larger range of values of $h$. For this test problem, the [NyR] method achieved 0.1% error in 390 s of walltime.

## 5.3. The motion of a torus sedimenting under gravity

As a final test case, we simulate the mobility problem of a torus sedimenting under the action of gravity (for detailed set-up and discretization, see appendix B.2). In the absence of an exact solution to this problem, we compare the distance travelled in the vertical direction after the system (equations (B6)–(B8) are solved for $t \in [0, 98.7]$. We compare the results obtained with the [Ny] and [NyR] methods with those from a simulation using the nearest-neighbour method ([NEAREST]) with a refined force discretization, disjoint force and quadrature discretizations and $\epsilon = 10^{-6}$. Figure 3*a–c* shows, respectively, example discretizations for the [Ny]/[NyR], and [NEAREST] methods, and the number of sDOF used in the [Ny] and [NyR] methods as $h$ is varied. For the [NEAREST] simulation, a highly resolved system is constructed with 14 667 sDOF and 231 744 quadrature points.

Figure 3*d–g* shows the convergence in $z$-position of the torus at $t = 98.7$ as both $\epsilon$ and $h$ are varied. The relative difference between these results and the [NEAREST] simulation are shown in figure 3*h–k*. The error behaves similarly to the previous cases: while [Ny] achieves accurate results with specific combinations of $\epsilon$ and $h$; by contrast [NyR] at relatively large values of $\epsilon = 0.1$–0.4 attains sub-1% error over an extended range of $h$ values.

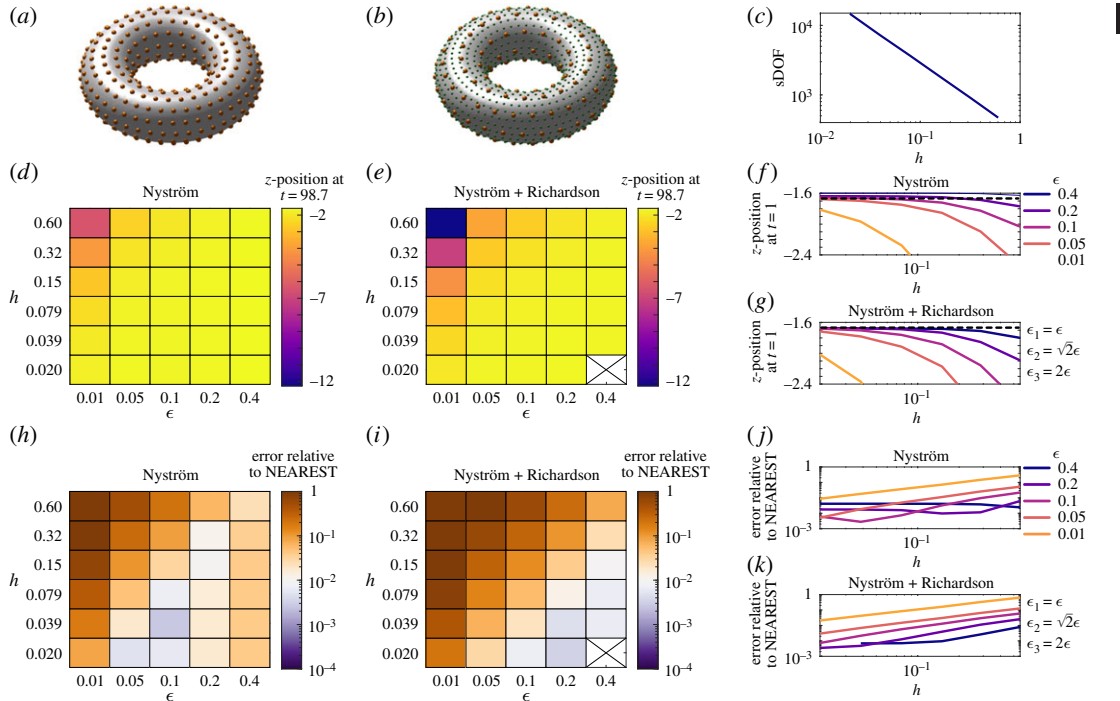

**Figure 3.** A torus, with central radius $R = 2.5$ and tube radius $r = 1$, sedimenting under gravity. (*a*) Sketch of the Nyström discretization (orange dots). (*b*) Sketch of the [NEAREST] force (large orange dots) and quadrature (small green dots) discretizations. (*c*) The number of scalar degrees of freedom used in Nyström and Nyström + Richardson calculations as $h$ is varied. (*d*) and (*e*) The $z$-position of the torus at $t = 98.7$ calculated with the Nyström and Nyström + Richardson methods as $\epsilon$ and $h$ are varied. (*f*) and (*g*) The same data plotted for each $\epsilon$ as $h$ is varied, with a dotted line showing the result using the nearest-neighbour method for comparison. (*h*) and (*i*) The error in $z$-position at $t = 98.7$ relative to the nearest-neighbour calculation. (*j*) and (*k*) The same data plotted for each $\epsilon$ as $h$ is varied. The cross in (*e,i*) denotes a parameter combination for which results could not be obtained due to near-singularity of the linear system.

As anticipated by the error analysis, the advantage of [NyR] appears in the range of relatively coarse values, i.e. $\epsilon = 0.1$–$0.4$. A solution could not be obtained when $\epsilon = 0.4$ and $h < 0.039$, due to the matrix system with $\epsilon_3 = 2\epsilon = 0.8$ becoming close to singular. For the choice of $\epsilon = 0.4$, the [NyR] method attained an error of 0.7% (compared with the result using [NEAREST]) in 144 s of walltime.

The results for the smallest choice of regularization parameter, $\epsilon = 0.01$, are not converged with $h$, consistent with our analysis in §3 focusing on moderate values of $\epsilon$ for which the quadrature error is subleading.

# 6. Discussion

This article considered the implementation of the regularized stokeslet method, a widely used approach in biological fluid dynamics for computational solution of the Stokes flow equations. An inherent challenge is the strong dependence of the degrees of freedom on the regularization parameter $\epsilon$, which necessitates an inverse-cubic relationship between the linear solver cost and the regularization parameter.

Here, we have investigated a simple modification of the widely used Nyström method, by employing Richardson extrapolation; performing calculations with three, coarse values of $\epsilon$ and extrapolating to significantly reduce the order of the regularization error. The method was compared with the original Nyström approach on three test problems: calculating the grand resistance matrices of the unit sphere and prolate spheroid, and simulating the motion of a torus sedimenting under gravity.

Investigation of these model problems has highlighted two significant phenomena, the first of which is well known but is worth repeating: (i) obtaining an acceptable level of error using the Nyström method is strongly dependent on being within the region where the (opposite-signed) regularization and quadrature errors exhibit significant cancellation, a phenomenon which has sensitive dependence on the discretization $h$ as $\epsilon$ is varied. (ii) The improvement in the order of regularization error provided by Richardson extrapolation is able to significantly and robustly reduce errors for simulations with

(relatively) large choices of $\epsilon$, enabling highly accurate results with relatively modest computational resources. This advantage is (by design) only maintained for these coarse values of $\epsilon$, so that the regularization error is subleading. Another approach which improves the order of convergence of the (important) local regularization error is given by Nguyen & Cortez [8], although the resulting regularized stokeslets may not be exactly divergence-free.

As discussed above, there are several existing approaches to improving the efficiency and accuracy of regularized stokeslet methods. The best approach in terms of strict computational complexity is the use of *fast* methods such as the kernel independent fast multipole method, which enables the approximation of the matrix-vector operation required for iterative solution of the linear problem [13,20], resulting in a $O(N \log N)$ method—although with somewhat greater implementational complexity. Another formulation is to borrow from the boundary element method developed for the standard singular stokeslet formulation [14], which has been applied to systems such as embryonic left-right symmetry breaking [21] and bacterial morphology [22]. The boundary element approach decouples the quadrature from the traction discretization and hence degrees of freedom of the system, enabling larger problems to be solved, although again at the expense of greater complexity through the need to construct a true surface *mesh*, with a mapping between elements and nodes. The nearest-neighbour discretization [15] retains much of the simplicity of the Nyström method, while separating the quadrature discretization from the degrees of freedom. Provided that the discretizations do not overlap, we still find this method to be an optimal combination of simplicity and efficiency. The Richardson approach does not avoid the need for the regularization parameter to not exceed the length scales characterizing the physical problem, for example the distance between objects. In this respect, the nearest-neighbour approach is advantageous because of its ability to accommodate smaller values of the regularization parameter.

In this work, we have focused on demonstrating how a numerically simple modification to the, already easy-to-implement, Nyström method can provide excellent improvements by employing coarse values of the regularization parameter $\epsilon$. This approach can be considered complementary to the nearest-neighbour method in its *coarse* philosophy and style: both methods are figuratively coarse in their simplicity, and literally coarse in their approach of increasing numerical parameters. The Richardson approach allows increases in the regularization parameter; the nearest-neighbour approach allows increase the force discretization spacing $h_f$. Either method enables more accurate results to be achieved with greater robustness and for lower computational cost. Moreover, both have the advantage of being formulated in terms of basic linear algebra operations, and therefore can be further improved through the use of GPU parallelization with minimal modifications [18]. The choice of which method to use is a matter of preference; the Richardson approach has the advantage of being immediately adoptable by any group with a working Nyström code, alongside the repeated calculations being embarrassingly parallel; the nearest-neighbour approach has the advantage of completely removing the dependence of the system size on $\epsilon$.

Accessible algorithmic improvements such as these provide the improved ability to solve a plethora of problems in very low Reynolds number hydrodynamics. Various potential application areas include microswimmers such as sperm [23,24], algae and bioconvection [25–29], mechanisms of flagellar mechanics [30,31], squirmers [32,33] and bio-inspired swimmers [34–36]. Stokeslet-based methods have been employed since the work of Gray and Hancock [6] in the 1950s; they continue to provide ease of implementation, efficiency and most importantly physical insight into biological systems.

Data accessibility. Data and relevant code for this research work are stored in GitLab: https://gitlab.com/meuriggallagher/the-art-of-coarse-stokes (MATLAB code for Nyström and Richardson extrapolation) and https://gitlab.com/meuriggallagher/NEAREST (MATLAB code for NEAREST and other dependencies); and have been archived within the Zenodo repository: https://doi.org/10.5281/zenodo.4632956.

Competing interests. We declare we have no competing interests.

Funding. This work was supported by Engineering and Physical Sciences Research Council (EPSRC) award no. EP/N021096/1. M.T.G. acknowledges support from EPSRC Centre grant no. EP/N014391/2.

Acknowledgements. We thank Eamonn Gaffney and Kenta Ishimoto for valuable discussion.

# Appendix A. Choice of extrapolation parameters

As a check on the robustness of the results presented in this article to the choice of extrapolation parameters ($\epsilon_1$, $\epsilon_2$, $\epsilon_3$), we calculate the relative error in the grand resistance matrix for the unit sphere (see §5) with the rules:

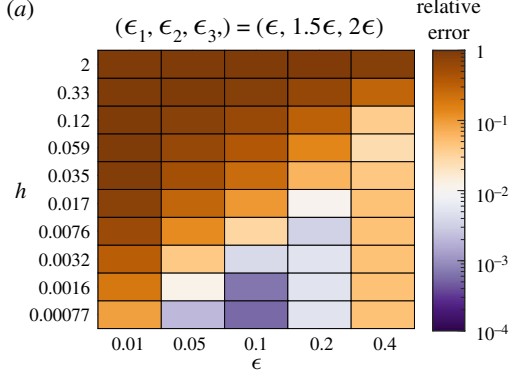
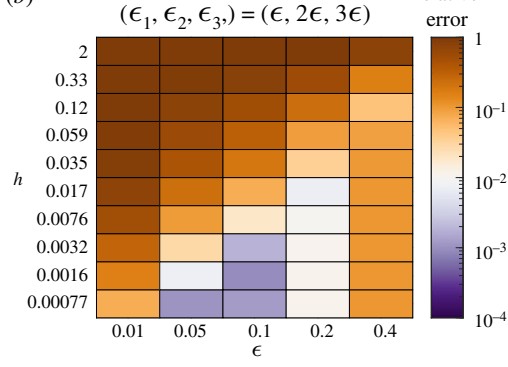
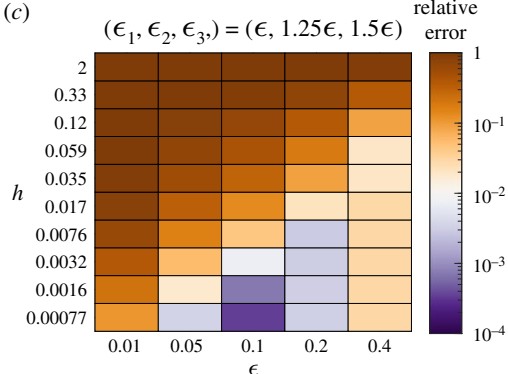
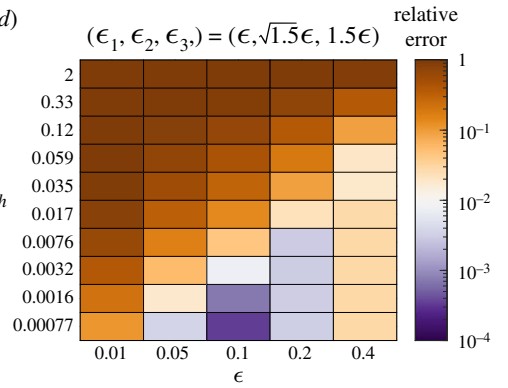

**Figure 4.** Relative error in calculating the grand resistance matrix for the unit sphere with the Nyström + Richardson method for four choices of extrapolation rule.

— $(\epsilon, \sqrt{2}\epsilon, 2\epsilon)$, figure 1;
— $(\epsilon, 1.5\epsilon, 2\epsilon)$, figure 4a;
— $(\epsilon, 2\epsilon, 3\epsilon)$, figure 4b;
— $(\epsilon, 1.25\epsilon, 1.5\epsilon)$, figure 4c;
— $(\epsilon, \sqrt{1.5}\epsilon, 1.5\epsilon)$, figure 4d.

Visual comparison between figures 1 and 4 shows that the improvement in accuracy is relatively similar.

# Appendix B. Further details of numerical experiments

## B.1. Discretization of the prolate spheroid

The location of points on the prolate spheroid, aligned with the $x$-axis, can be expressed in terms of the prolate spheroidal coordinates, as

$$x = \alpha \cosh \mu \cos \nu, \tag{B1}$$
$$y = \alpha \sinh \mu \sin \nu \cos \phi \tag{B2}$$
and
$$z = \alpha \sinh \mu \sin \nu \sin \phi, \tag{B3}$$

for $\nu \in [0, \pi]$, $\phi \in [0, 2\pi]$, with

$$\alpha = \sqrt{a^2 - c^2} \quad \text{and} \quad \mu = \arccos \frac{a}{\alpha}, \tag{B4}$$

where $a$ and $c$ are the major- and minor-axes lengths, respectively. We first discretize $\nu$ into $n$ uniformly spaced points, providing a discretization in $x$ which is slightly more dense in regions of higher curvature. For each choice of $\nu_i$ ($i \in [1, n]$), we discretize $\phi$ into $m_i$ linearly spaced points, where the choice

$$m_i = \left\lceil \frac{2\pi\alpha \sinh \mu \sin \nu_i}{h} \right\rceil, \quad i \in [1, n], \tag{B5}$$

ensures that each ring is approximately evenly discretized with spacing $h$. Here, $\lceil \cdot \rceil$ represents the ceiling function.

## B.2. A torus sedimenting under gravity

The equations of motion for a torus sedimenting under gravity are given by

$$- U_i - \epsilon_{ijk}\Omega_j(x_k - x_{0k}) - \frac{1}{8\pi}\iint_{\partial D} S_{ij}^\epsilon(\boldsymbol{x}, \boldsymbol{X})f_j(\boldsymbol{X})\, dS_{\boldsymbol{X}} = 0, \quad \forall \boldsymbol{x} \in \partial D, \tag{B6}$$

$$\iint_{\partial D} f_i(\boldsymbol{X})\, dS_{\boldsymbol{X}} = -1 \tag{B7}$$

and

$$\iint_{\partial D} \epsilon_{ikj}X_k f_j(\boldsymbol{X})\, dS_{\boldsymbol{X}} = 0, \tag{B8}$$

where repeated indices are summed over $i \in [1, 2, 3]$, $\boldsymbol{U}$ and $\boldsymbol{\Omega}$ are the translational and rotational velocities of the torus, $\partial D$ defines the surface of the torus, the central- and tube-radii of the torus are given by $R$ and $r$ respectively, and $\epsilon_{ijk}$ is the Levi-Civita symbol. The term on the right-hand side of equation (B7) derives from the (dimensionless) effect of gravity. The motion of the torus can be expressed as a system of nine ordinary differential equations for the time derivatives of the torus position $\boldsymbol{x}_0$ and basis vectors $\boldsymbol{b}^{(1)}$ and $\boldsymbol{b}^{(2)}$ (after which $\boldsymbol{b}^{(3)} = \boldsymbol{b}^{(1)} \times \boldsymbol{b}^{(2)}$). More details of how this 'mobility problem' is solved can be found in [15]. While this problem could be further constrained by enforcing that the angular velocity is zero (due to the symmetry of the torus), we focus on solving for the full rigid body motion. The mobility problem is solved using the [Ny], [NyR] and [NEAREST] methods, with results given in §5.3.

Points on the torus surface can be written as

$$x = (R + r\cos\theta)\cos\phi, \tag{B9}$$
$$y = (R + r\cos\theta)\sin\phi \tag{B10}$$

and

$$z = r\sin\theta, \tag{B11}$$

for $\theta, \phi \in [0, 2\pi]$. We discretize $\theta$ into $n = \lceil 2\pi r/h \rceil$ linearly spaced points, ensuring points on each ring are approximately evenly spaced with lengthscale $h$. For each $\theta_i$ ($i \in [1, n]$), we discretize $\phi$ into $m_i$ linearly spaced points via

$$m_i = \left\lceil \frac{2\pi(R + r\cos\theta_i)}{h} \right\rceil, \quad i \in [1, n], \tag{B12}$$

resulting in an approximately evenly spaced discretization for the torus with lengthscale $h$. For simulations with the [NEAREST] method, a fine quadrature discretization is created following the same process with lengthscale $h_q = h/4$. To ensure disjoint force and quadrature discretizations in this case, a filtering step is performed to remove any quadrature points which lie within a distance $h_q/10$ from their nearest force point.

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
