## [Peer Review File · Royal Society Open Science]

Review History

RSOS-210108.R0 (Original submission)

Review form: Reviewer 1

Is the manuscript scientifically sound in its present form?

Yes

Are the interpretations and conclusions justified by the results?

Yes

Is the language acceptable?

Yes

Do you have any ethical concerns with this paper?

No

Have you any concerns about statistical analyses in this paper?

No

Recommendation?

Accept as is

Comments to the Author(s)

One way to obtain regularisation error that is higher order in epsilon is to impose moment conditions on the singular kernels, i.e. the integral of the kernel times an appropriate power of x should be zero. I expect this could lead to comparable results. However, in case boundaries are close to each other, so that some integrals are nearly singular rather than singular, either strategy would be less likely to succeed.

Review form: Reviewer 2 (Richard Clarke)**Is the manuscript scientifically sound in its present form?**

Yes

Are the interpretations and conclusions justified by the results?

Yes

Is the language acceptable?

Yes

Do you have any ethical concerns with this paper?

No

Have you any concerns about statistical analyses in this paper?

No

Recommendation?

Accept with minor revision (please list in comments)

Comments to the Author(s)

Please see the attached (Appendix A).

Decision letter (RSOS-210108.R0)

Dear Dr Gallagher

On behalf of the Editors, we are pleased to inform you that your Manuscript RSOS-210108 "The art of coarse Stokes: Richardson extrapolation improves the accuracy and efficiency of the method of regularized stokeslets" has been accepted for publication in Royal Society Open Science subject to minor revision in accordance with the referees' reports. Please find the referees' comments along with any feedback from the Editors below my signature.

Please submit your revised manuscript and required files (see below) no later than 7 days from today's (ie 02-Mar-2021) date. Note: the ScholarOne system will 'lock' if submission of the revision is attempted 7 or more days after the deadline. If you do not think you will be able to meet this deadline please contact the editorial office immediately.

on behalf of Dr Peter Stewart (Associate Editor) and Mark Chaplain (Subject Editor)
openscience@royalsociety.org

Reviewer comments to Author:
Reviewer: 1

Comments to the Author(s)

One way to obtain regularisation error that is higher order in ϵ is to impose moment conditions on the singular kernels, i.e. the integral of the kernel times an appropriate power of x should be zero. I expect this could lead to comparable results. However, in case boundaries are close to each other, so that some integrals are nearly singular rather than singular, either strategy would be less likely to succeed.

Reviewer: 2

Comments to the Author(s)
Please see the attached

===PREPARING YOUR MANUSCRIPT===

Your revised paper should include the changes requested by the referees and Editors of your manuscript. You should provide two versions of this manuscript and both versions must be provided in an editable format:
one version identifying all the changes that have been made (for instance, in coloured highlight, in bold text, or tracked changes);

===PREPARING YOUR REVISION IN SCHOLARONE===

- If you are requesting a discretionary waiver for the article processing charge, the waiver form must be included at this step.
- If you are providing image files for potential cover images, please upload these at this step, and inform the editorial office you have done so. You must hold the copyright to any image provided.
- A copy of your point-by-point response to referees and Editors. This will expedite the preparation of your proof.

- Ensure that your data access statement meets the requirements at <https://royalsociety.org/journals/authors/author-guidelines/#data>. You should ensure that you cite the dataset in your reference list. If you have deposited data etc in the Dryad repository, please only include the 'For publication' link at this stage. You should remove the 'For review' link.
- If you are requesting an article processing charge waiver, you must select the relevant waiver option (if requesting a discretionary waiver, the form should have been uploaded at Step 3 'File upload' above).
- If you have uploaded ESM files, please ensure you follow the guidance at <https://royalsociety.org/journals/authors/author-guidelines/#supplementary-material> to include a suitable title and informative caption. An example of appropriate titling and captioning may be found at https://figshare.com/articles/Table_S2_from_Is_there_a_trade-off_between_peak_performance_and_performance_breadth_across_temperatures_for_aerobic_scope_in_teleost_fishes_/3843624.

Author's Response to Decision Letter for (RSOS-210108.R0)

See Appendix B.

Decision letter (RSOS-210108.R1)

Dear Dr Gallagher,

It is a pleasure to accept your manuscript entitled "The art of coarse Stokes: Richardson extrapolation improves the accuracy and efficiency of the method of regularized stokeslets" in its current form for publication in Royal Society Open Science.

At this stage, we ask that you please archive your GitHub code within the Zenodo repository: <https://guides.github.com/activities/citable-code/>. By doing this, a formal, citable DOI will be associated with your data record, and an open license (CC-BY preferred) can be applied to your data. We would then ask that you please update your data availability statement to read as:

"Data and relevant code for this research work are stored in GitHub: [GitHub URL here] and have been archived within the Zenodo repository: <https://doi.org/zenodo.....> [ref number].

You can expect to receive a proof of your article in the near future. Please contact the editorial office (openscience@royalsociety.org) and the production office (openscience_proofs@royalsociety.org) to let us know if you are likely to be away from e-mail contact – if you are going to be away, please nominate a co-author (if available) to manage the proofing process, and ensure they are copied into your email to the journal.

on behalf of Dr Peter Stewart (Associate Editor) and Mark Chaplain (Subject Editor)
openscience@royalsociety.org

Appendix A

The art of coarse Stokes: Richardson extrapolation improves the accuracy and efficiency of the method of regularized stokeslets by Gallagher and Smith applies Richardson extrapolation to a number of resistance problems formulated in boundary integral form using regularised stokeslets. Specifically, they consider rigid body motion of a sphere, prolate spheroid and sedimenting torus, benchmarking solution against non-extrapolated formulations, and nearest neighbour approaches which decouple discretisation of flow variables (traction) from quadrature discretisations.

I found the manuscript to be well written, and the favourable comparisons against the other approaches to be sound evidence of the merits of approach. As such I am supportive of publication.

There were a few areas, however, where I felt the manuscript might benefit from some expansion and/or clarification:

- i) The requirement that the $O(h^2/\epsilon)$ quadrature error be subdominant seems to be central to the analysis, but are there situations where this assumption might be problematic? Perhaps closely packed slender filaments? It might be useful if the authors could critique this assumption more in the Discussion section.
- ii) Related to the above point, it was not clear to me why the discretion error has minimal dependence on ϵ , and can be taken to be h^2/ϵ_0 ?
- iii) In the numerical experiments, the authors choose their extrapolation points to be $\epsilon, \sqrt{2\epsilon}, 2\epsilon$. Why these particular points, and is the accuracy of the Richardson extrapolation sensitive to this choice?
- iv) The authors report walltime for NyR, but I am wondering if it could be useful to compare walltime across all of the methods – since one of the disadvantages of Richardson extrapolation is that it is necessary to perform multiple computations. (Also, would CPU time be a more useful measure than walltime, since walltime varies with the hardware architecture?)
- v) In Figure 3g, it does not look as though the NyR simulation for smallest ϵ is converging to the correct z position. Perhaps the authors might comment on what is happening here?
- vi) In Appendix A(b), my intuition says that the torus should not be rotating given the geometry symmetry of the problem (would be interested to be corrected here), so is it not safe to assume the angular velocities and moments are zero? Also, is the term on the right hand side of (A.7) the Stokes drag on a torus?

More minor comments:

- a) In (1.8)-(1.10), should \mathbf{x} be $\hat{\mathbf{x}}$?
- b) I found the notation on the left hand side of (3.6) somewhat unusual, as earlier in the analysis M is written as a function of just one epsilon.
- c) Line 46, p10: Should 1c read 2c?
- d) It could be useful to formally define the norm used in (5.1).
- e) I must admit to not fully understanding the source of contained and disjoint errors in the nearest neighbour approach. If the traction

discretisations are decoupled from the quadrature, why is the quadrature error a function of the traction collation point?

Appendix B

The art of coarse Stokes: Richardson extrapolation improves the accuracy and efficiency of the method of regularized stokeslets

List of revisions

M.T. Gallagher and D.J. Smith

Please find in this document our point-by-point responses to the reviewers, and the complete list of revisions. In each response the page numbers refer to the version with highlighted changes (additions in blue, removals in red).

In addressing a point made by Reviewer 2, we noticed that the expression used on the right-hand side of equation (A7) was unnecessarily complicated, as the force magnitude can be scaled out of the problem, so that the right-hand side is -1 . This change has no substantive effect on the results other than an effective change to the (arbitrary) time interval used for the analysis, which has been modified in section 5c.

Reviewer 1

One way to obtain regularisation error that is higher order in epsilon is to impose moment conditions on the singular kernels, i.e. the integral of the kernel times an appropriate power of x should be zero. I expect this could lead to comparable results. However, in case boundaries are close to each other, so that some integrals are nearly singular rather than singular, either strategy would be less likely to succeed.

We thank the reviewer for their comments surrounding other ways to reduce regularisation error. We have added discussion surrounding this point on page 2, as well as page 11, with reference to Nguyen and Cortez [8].

Reviewer 2

We thank the reviewer for their comments regarding this work, and for the suggestions of expansion and clarification. We believe we have been able to address each of the reviewer's comments and have made modifications as follows:

i) The requirement that the $O(h^2/\epsilon)$ quadrature error be subdominant seems to be central to the analysis, but are there situations where this assumption might be problematic? Perhaps closely packed slender filaments? It might be useful if the authors could critique this assumption more in the Discussion section.

We agree and have added to the discussion (page 11) to clarify that the Richardson approach does not avoid the need to the regularisation parameter to not exceed the length scales characterising the physical problem, for example in the case of slender filaments.

ii) Related to the above point, it was not clear to me why the discretion error has minimal dependence on ϵ , and can be taken to be h^2/ϵ_0 ?

We have clarified the dependence of the individual error terms on the parameters h and ϵ at the start of section 3 (page 4). In short, for a choice of regularisation parameter that is not excessively small, the quadrature error h^2/ϵ is subleading to the force discretisation error (h), and hence the full error associated with discretisation (the sum of the force and quadrature errors) has minimal dependence on ϵ .

iii) In their numerical experiments, the authors choose their extrapolation points to be $(\epsilon, \sqrt{2}\epsilon, 2\epsilon)$. Why these particular points, and is the accuracy of the Richardson extrapolation sensitive to this choice?

Thank you for this suggestion. We have now added Appendix A, within which we compare a number of rules, demonstrating that the accuracy of the method is not overly sensitive to the choice of parameters.

iv) The authors report walltime for NyR, but I am wondering if it could be useful to compare walltime across all of the methods - since one of the disadvantages of Richardson extrapolation is that it is necessary to perform multiple computations. (Also would CPU time be a more useful measure than walltime, since walltime varies with the hardware architecture?)

Due to the embarrassing parallelism of the required calculations, the [NyR] method is, at worst, three times the cost of the [Ny] method on any hardware, with additional savings possible through the use of parallel computing architecture. We have added clarification about this at the end of section 3 (page 5). We presented a walltime result rather than CPU time, as the latter does not take into account important features such as the cost of memory transfer.

v) In Figure 3g, it does not look as though the NyR simulation for smallest ϵ is converging to the correct z position. Perhaps the authors might comment on what is happening here?

We agree with the reviewer that this requires additional comment (see our response to point i) above). We have added explanation at the end of section 5c (page 9). Briefly, for excessively small choices of ϵ the dominant error term is the quadrature error, not the regularisation error, negating the benefits of the Richardson approach.

vi) In Appendix A(b), my intuition says that the torus should not be rotating given the geometry symmetry of the problem (would be interested to be corrected here), so is it not safe to assume the angular velocities and moments are zero? Also, is the term on the right hand side of (A.7) the Stokes drag on a torus?

The reviewer's intuition is correct, and we have commented on this in (what is now) Appendix B(b). We have constructed the sedimenting torus problem as a 'mobility problem', wherein the total moment on over the surface is zero, the total force is a prescribed non-zero value, and we solve for the full rigid body motion of the torus. We thank the reviewer for pointing out that the right hand side of A7 can be scaled to (-1) with a change of time-scale, which results in slight modification to the time interval represented by the results in section 5c.

Minor comment a) In (1.8)-(1.10) should \mathbf{x} be $\hat{\mathbf{x}}$?

The notation is correct, the hat for $\hat{\mathbf{e}}_k$ is because it is a unit vector.

Minor comment b) I found the notation on the left hand side of (3.6) somewhat unusual, as earlier in the analysis M is written as a function of just one ϵ .

The notation on the LHS of 3.6 is to highlight the dependence on (the now) three values of extrapolation parameters, as opposed to the single regularisation parameter previously.

Minor comment c) Line 46, p10: Should 1c read 2c?

We have corrected this typographical error.

Minor comment d) It could be useful to formally define the norm used in (5.1).

We have clarified that this is the 2-norm.

Minor comment e) I must admit to not fully understanding the source of contained and disjoint errors in the nearest neighbour approach. If the traction discretisations are decoupled from the quadrature, why is the quadrature error a function of the traction collocation point?

We have now clarified on page 6 that the 'quadrature error' specifically refers to the quadrature error that occurs when performing collocation on the traction points (hence the dependence on the traction point location).

A full list of corrections is provided at the end of this document.

Full list of corrections

- P1 - Resistance problem -> resistance and mobility problems;
- P2 - Added citation for Cortez et al. 2005;
- P2 - Added discussion and reference around improvements to the near-field convergence;
- P3 - Added explanation of how the mobility problem is solved;
- P4 - Clarified that choice of ϵ ensures quadrature error is subleading, and why the discretisation error has minimal dependence on ϵ in this case;
- P5 - Highlighted that the [NyR] method is, at most, three times the cost of the [Ny] method;
- P6 - Clarified that the quadrature error is associated with collocation;
- P7 - Highlighted that the choice of extrapolation rule is now discussed further in Appendix A;
- P7 - Provided formal definition of the choice of matrix norm;
- P8 - Figures 1c -> Figures 2c;
- P8 - Figures 1d -> Figures 2d;
- P8 - Corrected order of results for error plateaus;
- P9-10 - Corrected time range for sedimenting torus simulations;
- P10 - Added discussion about the results for a sedimenting torus with $\epsilon = 0.01$;
- P11 - Added discussion surrounding the order of convergence of the local regularisation error;
- P11 - Added discussion around the choice of method for small ϵ ;
- P12 - Added comment that the repeated calculations for the [NyR] method can be parallelised;
- P12 - Added additional funding acknowledgment;
- P12 - Added appendix discussing choice of extrapolation rule;
- P14 - Changed time scaling in equations A6-A8 to simplify equation A7;
- P14 - Added discussion surrounding additional constraints on the sedimenting torus simulation;
- P15 - Added citations: Nguyen and Cortez 2014, and Smith 2018.